# Exploring the Role of Deep Learning Technology in the Sustainable Development of the Music Production Industry

**Sung-Shun Weng [1] and Hung-Chia Chen [2],*** 

[1] Department of Information and Finance Management, National Taipei University of Technology, 1, Sec. 3, Zhongxiao E. Rd., Taipei 10608, Taiwan; wengss@ntut.edu.tw

[2] College of Management, National Taipei University of Technology, 1, Sec. 3, Zhongxiao E. Rd., Taipei 10608, Taiwan

\* Correspondence: honga.chen@gmail.com; Tel.: +886-2-2771-2171 (ext. 6500)

**Abstract:** This study explores the role of deep learning technology in the sustainable development of the music production industry. This article surveys the opinions of Taiwanese music creation professionals and uses partial least squares (PLS) regression to analyze and elucidate the importance of deep learning technology in the music production industry. We found that deep learning cannot replace human creativity, but greater investment in this technology can improve the quality of music creation. In order to achieve sustainable development in the music production industry, industry participants need to awaken consumers' awareness of music quality, actively enhance the unique value of their art, and strengthen cooperation between industries to provide a friendly environment for listeners.

**Keywords:** music production industry; music-art; deep-learning technology; music for sustainability

## 1. Introduction

Human beings have the capacity to create art, which reflects life, thoughts, and emotions. Music is the art that uses the sense of hearing to free souls, convey feelings, and disseminate cultures [1–3]. Distinct from the audience facing the music market, the music production industry emphasizes the artistic creation and performance of music.

The existence of music in different eras and regions reveals its importance to human life. In recent years, our daily lives have been significantly influenced by the development and innovation of technology. Indeed, the use of technology affects the development of industry. For example, enterprises implement technologies to hasten production, reduce costs, and create better value. The music production industry also takes advantage of digital technology. Since the release of the MIDI (Musical Instrument Digital Interface) standard in 1983, artificial intelligence [4] has been envisioned as the next key technology to have a large impact on the music production industry. In order to sustain development, the music industry has experienced changes in its industrial structure caused by technology and undergone corporate restructuring, restructuring of its industrial chain, design changes of its copyright mechanisms, transformations of its channels, and interest sharing disputes. Each impact also affects the model of the music production industry, especially as it relates to revenue and consumer perception.

Deep learning, a hot topic in artificial intelligence [5,6], is applied in many projects, including automatic processing and the recognition of texts, languages, voices, images, and so on. Some deep learning projects focus on music production [7], such as the identification of music scores, automatic

composition, and style recognition [8–13]. For example, the Google Magenta [9,14–16] project enables spontaneous interaction between a computer and a pianist. Due to the great efforts of worldwide scholars and professionals in recent years, deep learning has successfully mastered various features and styles of past music pieces and automatically created music to satisfy audiences [17]. Deep learning is recognized as an innovative field that could help the music production industry change and evolve in the future [18].

However, the topic of whether deep learning will replace music creation, resulting in changes in music production modes, the loss of job opportunities, and impacts on the sustainable development of the entire industry, has raised concerns among music production staff. To face these future challenges, the industry needs to develop new sustainability strategies.

With the rise of consumer awareness in the digital economy and consumer preferences for hedonic products and services, it is obvious that consumer taste plays an important role in judgment and decision-making. In addition to technology, the music production industry should consider external factors such as the artistic values of music and kitsch culture [3,19]. To sum up, the value chain ideology of the music production industry and trends of the customer market play essential roles in final music products [20]. As Stefan and other scholars point out, stakeholders should maintain the best balance between working artistically and engaging in business concerns, the so called, "sweet spot" [21], as well as find the best coexistence model for art and business in order to create corporate value [22].

The first part of this study introduces the music production industry and its main work. The second part explores past scholars' views on music presentation and artistic creation and quality to understand the measurement factors needed for music production. The third part discusses the current results of music creation experiments through deep learning techniques to extract the elements of music. The fourth and fifth sections, respectively, discuss the use of technology and the satisfaction of art products and integrate the above-mentioned literature design, research structure, and hypothesis. The third section and the rest of the paper analyze the research hypothesis and propose management suggestions by investigating the opinions of music production professionals.

This study explores the nature of music and its relationship to the quality of music production and music producers. Then, this study proposes the future of sustainable development of the music production industry.

## 2. Literature Discussion

### 2.1. Music Production Industry

Music production is part of the music industry. The music industry covers music creation, music production, agencies, copyright, distribution, marketing, and retail or online channels [23]. In addition, the music industry connects performers with consumers [24].

Since the appearance of MP3, MPEG (Movie Picture Experts Group) 1 Layer 3, a digital audio encoding and distortion compression technology, and peer-to-peer networks (P2P) technologies from 1998 to 1999, the music industry has suffered from piracy and copyright issues. Legal protections did not evolve as quickly as the application of new technology [25]. Digital technology has continued to evolve into online audio streaming technology [26] and cloud digital locker [27] technology, which bring convenience to consumers. However, the music industry needs to adjust itself to this new technology, especially in the areas of copyright protection and business loss and profit, which will affect the sustainable development model of every enterprise in the industry chain [28,29].

Distinct from the demands of the music industry, the music production industry pays more attention to the artistic elements of music, as well as performance and auditory presentation. Music production includes instrumental performance, sound recording, and music editing. The industry chain includes music creation, composition, arrangement, instrumental performance, recording, mixing, and editing. The final musical works are distributed on tapes, CDs, vinyl records, digital files, and/or online streaming, or used in the fields of video shows, games, and movies. Earlier, music production

was completed via analog tape players, analog sound sources, and analog samplers, and later via digital music tools [30,31]. Digital technology has lowered the threshold to enter the field of music production. Through digital music tools, the quality of music arrangement, recording, and mixing has also improved. Music producers do not need to judge the quality of music based on their own senses because the digital tools help achieve a balanced music performance that also retains the uniqueness of the music producer. Digital technology continues to evolve, and artificial intelligence technology will certainly affect and challenge the most unique emotional performances of human beings.

Taiwan senior musician who has been engaged in the music production industry for more than 20 years, said: "At present we are increasingly relying on the application of technology, and we also look forward to the convenience and change." There is no doubt that the music production process has involved the use of many technologies and greatly improved production capacity. The Expert also said: "We are also very aware that the technology of artificial intelligence deep learning cannot replace all production processes and human unique ideas, but it may cause changes in music perception to consumers and affect the relationship of the industry." It can be seen that the application direction of technology, music presentation, consumer perception, market mechanism, industrial value chain, etc., are all topics that need to be explored in the current sustainable development of the music production industry. The industry chain relationship between the music industry and the music production industry is shown in Figure 1.

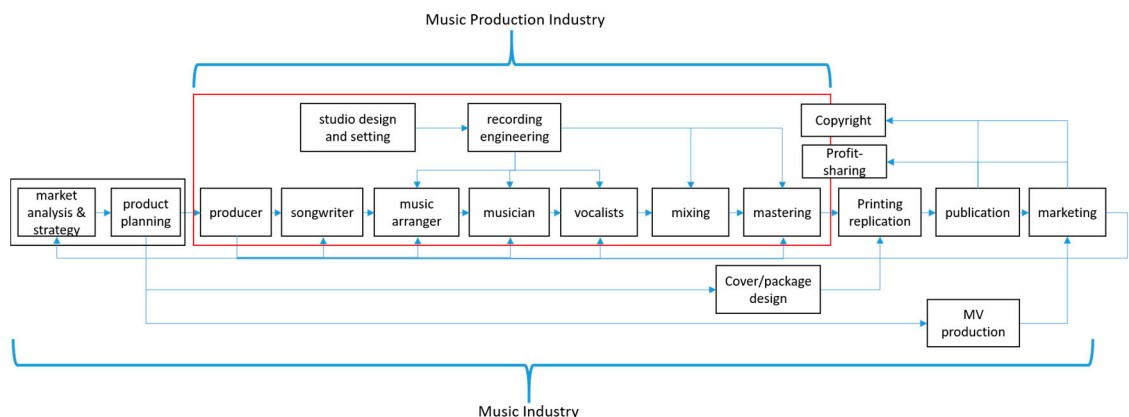

**Figure 1.** Music industry and music production industry structural diagram.

## 2.2. Music as Artistic Expression

Art is an organic product of human knowledge, emotions, ideals, experiences, and concepts. Music is an art of time while visual work is an art of space [1,3]. The beauty of music is sensed through sound and implemented by activities of external acoustic stimulation and internal auditory experience [32]. Distinct from other forms of art, music not only affects the listeners' thoughts but also satisfies their spiritual needs [1]. Music is an art that uses the sense of hearing to free souls, convey feelings, and disseminate cultures.

Music is a method of expressing ideas to listeners through the medium of sound. The elements of music include rhythm, melody, harmony, timbre, form, texture, and dynamics. The scope of music includes various combinations of ambient sound, percussion, special effects, and even noises. Through well planned and organized designs, the use of technology, and the integration of the creators' emotions and vitality, touching music pieces are composed. This process is known as the essence of music. Rayfrey once said that music is a product of the mind that combines external sound and the inner spirit. When the two are united, it is called "tone". When the external and the internal elements are the same and in harmony, "music" is produced [33]. Li Qingzhu [34] described how music represents a language of the spiritual world that depicts the state of the soul. Music is like an auditory image composed of musical notes, containing the rich emotional language that the musician wants to

express, as well as the concepts of aesthetic thought and culture [35]. Webster proposed that, when engaged in music production, producers, based on their comprehension of musical elements, should achieve their creative goals through personal motivation, character, maturity, their social/cultural environment, schoolwork, interpersonal relationships, and past experiences—so-called "creation conditions". Producers should also engage in a continuous diffusion and convergence of thinking and understanding of concepts, techniques, and aesthetic perceptions—so-called "skill strengthening" conditions [36]. Personality also affects the process of music creation, production, and presentation. Psychologists describe personality traits using five broad dimensions (the Big Five model), including extraversion, neuroticism, agreeableness, conscientiousness, and openness to experience. They believe that personality is gradually developed and formed under the interaction of heredity and environment. Personality does not only indicate the outer appearance and behavior of a person but also includes the morals, beliefs, and self-perceptions of a person, also known as character [37].

Art belongs to the emotional and spiritual level of human experience and is difficult to quantify. However, we can learn how to appreciate art by understanding its elements. Musical creativity is expressed through the various elements of music, including melody, rhythm, timbre, and sound quality, to draw the lines and shapes of music; harmony fills in the color and space of music; texture, strength, and speed represent the structure of music; and form is the performance of music. The image of music is not only a process of creativity but also the performance and cooperation of elements and factors that trigger different feelings among the audience because musical communication involves the perceptual process of performance and external coordinators [38]. Further, listeners' self-cultivation levels and tastes affect their artistic experiences even at the same concert [39]. Interactions between different audiences and performers also affect performance, production, and listening experience [40].

The aesthetics of music also depend on many external factors, such as post-capitalism and profit-driven economies and kitsch culture [41]. Secular culture, commercialization, technology, interactions between people and media, and live concert performances have become the hottest topics in discussions on postmodern popular music [42]. Even innovative music pieces can be possibly transformed from the kitsch formula to generate originality [3].

This phenomenon is especially observed when postmodernism or futurism challenges the traditional music model by producing electronic sounds, dance music, rock, mixed mashups, new century music, and so on. Futurism allows composers to go beyond traditional music materials and manipulate various natural sounds and noises, which results in the link between new technologies and the manifestations of musical material [43].

Through literature, we see that the quality of music production is significantly affected by the technical skills, emotional personality, and external interactions of the producer. The quality of music also affects the appropriateness of the music's presentation. Drawing upon this literature, this study proposes the following hypotheses:

**Hypothesis 1 (H1).** *Music production staff's "techniques and capabilities for music production" have a positive and significant effect on the "quality of music production".*

**Hypothesis 2 (H2).** *Music production staff's "emotions and feelings for music production" have a positive and significant effect on the "quality of music production".*

**Hypothesis 3 (H3).** *The "external interacting factors affecting music production" have a positive and significant effect on the "quality of music production".*

**Hypothesis 4 (H4).** *The "quality of music production" has a positive and significant effect on "showing appropriateness".*

### 2.3. Deep Learning in the Field of Music Production

In 2004, Geoffrey Hinton, a professor in the field of artificial neural networks (ANNs) at the University of Toronto, renamed artificial neural networks as "deep learning". Through the simulation of biological multi-layer neural networks, scholars have developed many layers, architectures, and initializations of artificial neural networks, such as the convolutional neural network (CNN) [44], recurrent neural network (RNN) [45,46], and long short-term memory (LSTM) [47], over the past 30 years. This study concentrates on deep learning in the field of music production.

In 1998, Hörnel and Menzel explained how a hybrid music-harmonization system was calculated by the feedforward network algorithm HARMONT [48] and MELONET [49] in a multiscale neural network model. This system analyzed Bach's music through music symbols and neural-like algorithms, capturing the features of the music, eventually automatically producing four choruses and some melodies of a similar style and predicting the melodies [12].

Later, in 2002, Douglas Eck and Jürgen Schmidhuber successfully completed the composition of Blues music through the LSTM (long short-term memory) algorithm. The first part of the study confirmed that LSTM, without relying on melody, could easily learn the structure of chords and yield a new piece of music; the second part showed that LSTM could learn the structure of chords and melody. Then, this structure was used to generate a new song.

In 2012, Boulanger-Lewandowski et al. used a piano roll and RNN chord music symbol sequence with a high-dimensional time distribution to analyze a MIDI file collection and applied this sequence to chord music generation and music editing [8,50].

In 2016, Gaetan Hadjeres and Francois Pachet proposed a DeepBach model to mimic the four-part chorale by Johann Sebastian Bach [8]. This experiment claimed to be successful, as music experts voted for the system to produce compelling and coordinated music in the style of Bach, although the authors did mention existing problems of plagiarism.

In 2017, Iman Malik proposed bi-directional LSTM with memory gating developed through the concept of a bi-directional RNN [51]. Each note velocity was encoded into its corresponding (pitch, timestep) index. Iman Malik finally confirmed that his musical model could produce human-like performances based on the evaluation of two Turing tests [13].

At present, Google Magenta provides a computer program that allows users to create melodies. Google uses Tensorflow, a large heterogeneous machine learning operating system [52] and trains the program models to generate music via the Melody RNN model using a user-friendly MIDI interface to interactively engage with it [15].

As a result of the research efforts of many scholars over the years on deep learning, a large scale of music can be analyzed for its distribution of notes (in staves, piano rolls, digital matrixes, and MIDI numbers), dynamics, and time series. Deep learning also suggests how music should be played and, therefore, creates a musical piece with excellent quality [17].

Throughout the literature, scholars note that deep learning technology can generate appropriate high-quality music content by analyzing various elements in music. Drawing upon this literature, this study proposes the following hypotheses:

**Hypothesis 5 (H5).** *"Deep learning" has a positive and significant effect on the "quality of music production."*

**Hypothesis 6 (H6).** *"Deep learning" has a positive and significant impact on "showing appropriateness".*

**Hypothesis 7 (H7).** *"Deep learning" has a positive and significant effect on the "satisfaction of music products."*

**Hypothesis 8 (H8).** *"Deep learning" has a positive and significant effect on the music production staff's "techniques and capabilities for music production".*

## 2.4. Switching Costs of Technology Usage

B. Jackson said, "the larger and more disruptive the investment actions required, the greater the customer's reluctance to change commitments and incur switching costs" [53]. This phenomenon is called a "switching barrier". The switching of technology certainly creates impacts; therefore, customers will have to evaluate related benefits and costs, such as money, time, and the uncertainties of changes [54], while music producers consider the quality of music and their psychological acceptance. For example, producers who prefer analog music will remain dissatisfied with and skeptical about digital technology. Burnham et al. identified three switching costs, including time-related procedural switching costs, money-related financial switching costs, and emotion-related relational switching costs [55]. Chen Huihuang and others mentioned in their research that the psychological factor of customers plays an important role in the switching cost model, followed by the procedural economic risk cost, assessment cost, learning cost, and relational consumer comfort and consumer approval [56]. Therefore, Hypotheses 9 and 11b established in this study indicate that the cost of the technology used will affect the quality of music production. This also has a mediating effect on the quality of music production and showing appropriateness.

**Hypothesis 9 (H9).** *The "switching cost of technology usage" has a significant and positive effect on the "quality of music production".*

## 2.5. Satisfaction of Music Products

Music products yield varying levels of customer satisfaction due to the unique sensory experience and feelings of each customer. Music is also a product of culture. The five major characteristics of cultural goods are: (1) abstractness, (2) subjectivity, (3) non-utilitarianism, (4) uniqueness, and (5) entirety [57]. Some researches connect customer satisfaction to cognition [58,59]. Therefore, showing the appropriateness of music, under the framework of the five characteristics of cultural goods, will affect customer satisfaction. In addition, artists also employ aesthetic points of view to illustrate the perception and experience of art goods [60,61]. Aesthetic experience is the process of experiencing beauty, which generates a sense of pleasure on a self-directed and self-centered basis [62]. The value of tangible products or services that customers receive, therefore, reflects an invisible experiential perspective of consumption, where the pleasure generated exceeds the time and money invested [63]. Mathwick et al. believe that in the process of consumption, the experiential value can affect customers' preferences and attitudes toward purchasing products; consequently, companies are motivated to utilize the experiential value model for marketing purposes. Based on Holbrook's experiential aspect of consumption, Mathwick et al. established an experiential value scale including playfulness, aesthetics, customers' "return on investment", and service excellence [64]. This study establishes Hypothesis 10, which indicates that the satisfaction and showing appropriateness of music products have significant effects.

**Hypothesis 10 (H10).** *"Showing appropriateness" has a positive and significant effect on the "Satisfaction of music products".*

**Hypothesis 11a (H11a).** *"Deep learning" has a mediating effect on both "quality of music production" and "showing appropriateness".*

**Hypothesis 11b (H11b).** *The "switching cost of technology usage" has a mediating effect on both the "quality of music production" and "showing appropriateness".*

## 3. Research Architecture

The research architecture was constructed based upon a comprehensive review of previous studies and research findings. Figure 2 illustrates the research architecture developed for this study.

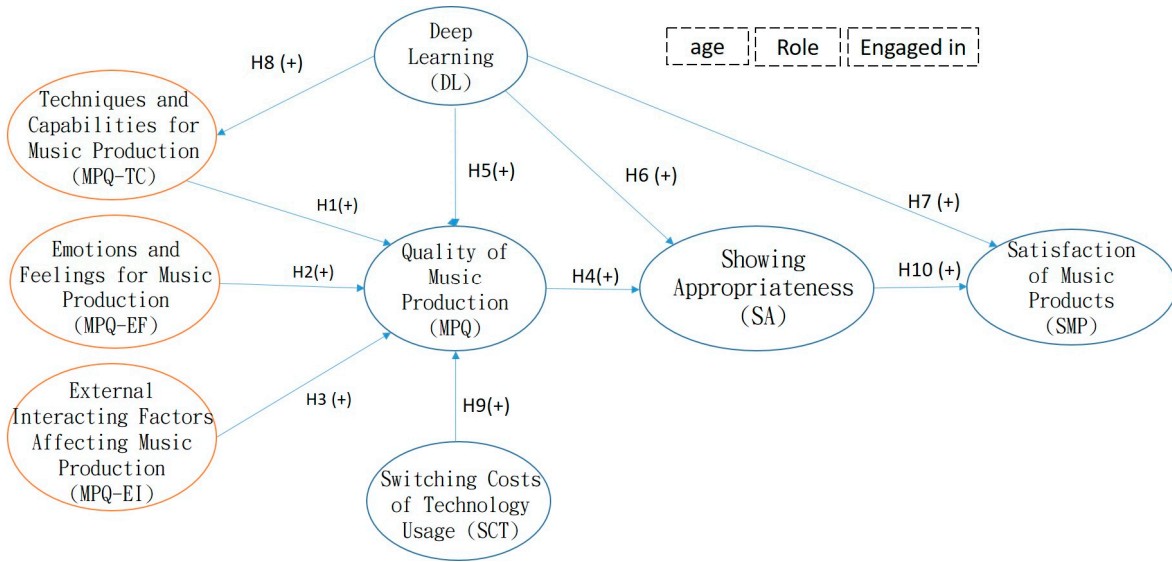

**Figure 2.** Research architecture.

Variables and operational definitions: An operational definition explains the terms of a process needed to decide the nature of a variable and its properties. To achieve reliability and validity, this study cites established questions and develops questions based on domestic or overseas studies discussed in Section 2. However, the constructs of "quality of music production" and "deep learning" are newer topics, so their questions are self-designed with reference to the literature discussed in Section 2. This study adopts a five-point Likert scale with variables, operational definitions, and references, as shown in Table 1.

**Table 1.** Variables, operational definitions, and references.

| Variables | Operational Definition | References |
|---|---|---|
| MPQ | Musical creativity is based on personal motivation, personality, maturity, social/cultural environment, schoolwork, interpersonal relationships, past experiences, and other "creative conditions" characteristics, as well as "strengthening skills", such as the comprehension of concepts, techniques, and aesthetics to achieve the quality of music production. | [36] |
| MPQ-EF | Emotions occur with the participation of cognition (brain regions), behavior, and the environment. Senses of affluence or poverty and contentment or desire, will affect the production staff's rational and emotional state and willingness to accept innovative ideas or confine them to conservative thinking as self-generated influences; for example, emotions determine behavior in the same way that external sources of influence do. | [35,65] |
| MPQ-TC | The value of music as a human spiritual experience is accomplished through the production staff's techniques and capabilities to master various elements of music so that the final products can represent the staff's emotions and vitality. | [2] |

**Table 1.** *Cont.*

| Variables | Operational Definition | References |
|---|---|---|
| MPQ-EI | External factors, such as live concerts and the cooperation of various roles, influence how the audience feels about the music. In addition, the self-cultivation levels and tastes of the audience result in different feelings for the same musical performance. | [38–40] |
| SA | Musical aesthetics are influenced by external factors, including post-capitalism, a profit-oriented market, kitsch culture, and so on. Therefore, "showing appropriateness" means that despite the manipulation of the commercial system, musical works can still arouse strong emotions from, and resonate with, the audience. | [3,38] |
| SMP | Satisfaction of music is the evaluation of consumption rewards, purchase costs, and expectations. | [59] |
| DL | Deep learning refers to the machine learning of past musical features to simulate and generate the emotional expression required in a specific type of music. | [11,13,18] |
| SCT | This term defines the customer's evaluation of the benefits and costs of switching to new technologies, services, or products. | [54,55] |

Notes: MPQ: quality of music production, MPQ-EF: emotions and feelings for music production, MPQ-TC: techniques and capabilities for music production, MPQ-EI: external interacting factors affecting music production, SA: showing appropriateness, SMP: satisfaction of music products, DL: deep learning, SCT: switching costs of technology usage.

## 4. Research Design

### 4.1. Measurement

To validate the model (Figure 2) and its hypotheses, this study disseminated a questionnaire survey among the production staff in the music industry in Taiwan. This study employed a survey instrument drawing upon a comprehensive literature review. To test the difficulty level of the questions, together with the reliability and validity of the scales, a pilot study was conducted in advance with a sample size of 50 music production staff to avoid errors in the questionnaire. Some questions were removed to reduce ambiguity and simplify interpretation after the feedback from the pilot test. The survey instrument and measurement items are exhibited in Appendix A.

### 4.2. Data

This survey was done in 2017 through online software. To ensure the quality of the data, three criteria were applied to the profiles of the survey respondents: comprehensive knowledge of music production, having a role in the music industry for more than five years, and continued research in music related fields. There were 105 valid responses.

As exhibited in Table 2, the sample comprises five different music staff roles, of which more than half are audio recording and mixing engineers (52.4%). Both audio recording and mixing engineers (52.4%) and music arrangers (43.8%) represent the majority of the sample. The majority of the sample worked in the music production field for less than 10 years (41%). Since the data were collected from a single source, for validity, common method bias was assessed. This study used Harman's post hoc single-factor analysis, a factorial analysis of all indicators was conducted, and the first extracted factors indicated 26.3% variance. This analysis confirmed that common method bias was unlikely to be an issue in the data for this study [66].

**Table 2.** Sample profile.

| Sample Characteristics (*n* = 105) | Obs. | (%) | Sample Characteristics | Obs. | (%) |
|---|---|---|---|---|---|
| **Roles in the Music Production Industry (Multiple Choice)** | | | **Age** | | |
| Music Creator | 42 | 40.0% | Less than 20 years old | 11 | 10.5% |
| Singer | 38 | 36.2% | 20–30 years old | 25 | 23.8% |
| Music Producer | 36 | 34.3% | 30–40 years old | 28 | 26.7% |
| Music Arranger | 46 | 43.8% | 40–50 years old | 30 | 28.6% |
| Audio Recording and Mixing Engineer | 55 | 52.4% | More than 50 years old | 11 | 10.5% |
| **Years in the Music Production Field** | | | **Sex** | | |
| No Experience | 6 | 5.7% | Male | 79 | 75.2% |
| Within 10 Years | 43 | 41.0% | Female | 26 | 24.8% |
| 10–20 Years | 18 | 17.1% | | | |
| 20–30 Years | 23 | 21.9% | | | |
| Above 30 Years | 15 | 14.3% | | | |

Notes: Data collected from Taiwan music production industry staff. This study used Harman's post hoc single-factor analysis. A factorial analysis of all indicators was conducted, and the first extracted factors indicated 26.3% variance.

## 5. Results

The partial least squares (PLS) method [67] was adopted for the estimation process of this conceptual model. The PLS method fulfils the purpose of this research by examining the validity of the constructs without requiring normally distributed variables. Next, PLS requires a sample size ten times the number of the largest number of structural paths directed at a particular construct [68]. In this conceptual model, the largest number of structural paths directed at a particular construct was 3, indicating that the minimum sample size was 30. This study had a sample size of 105 (*n* = 105), which was adequate for PLS. Before testing the structural model, this study investigated the measurement model to assess reliability and validity.

### 5.1. Measurement Model

This study examined indicators of reliability, construct reliability, convergent validity, and discriminant validity in order to assess the measurement model. Tables 3 and 4 present the results of the measurement model. The indicator of reliability is considered based on loadings above 0.7 and formative indicator loading weights above 0.2. This study designed techniques and capabilities for music production (MPQ-TC), emotions and feelings for music production (MPQ-EF), and external interacting factors affecting music production (MPQ-EI) constructs as formative indicators. Due to collinearity, the variance inflation factor (VIF) of each question needed to be less than 10. Therefore, nine items (MPQ-TC1, MPQ-EI3, deep learning (DL)4, DL7, DL8, showing appropriateness (SA)3, switching costs of technology usage (SCT)5, SCT6, and SCT7) were omitted. Table 4 indicates the instruments with good indicator reliability where the loadings are above 0.7, except for the formative indicators of MPQ-EF2, whose loading weights are less than 0.2. The composite reliability coefficient assesses the construct reliability, which considers indicators of different loadings [67,69].

Table 4 reveals that each construct's composite reliability is above 0.7, showing the reliability of the constructs. To determine the convergent validity, this study used the average variance extracted (AVE), with an acceptable value higher than 0.5 (i.e., the latent variable explains more than half of the variance of its indicators) [69,70]. Table 5 indicates that each construct's AVE meets this requirement. For discriminant validity assessment, this study adopted two dominant approaches: the Fornell–Larcker criterion and the examination of cross-loadings. According to the Fornell and Larcker testing system [70], first, the levels of the square root of the AVE for each construct should be greater than the correlation involving the constructs (as shown in the Table 5 with AVEs in bold). Second, the loading of each indicator should be greater than all cross-loadings [71] (shown in Table 4). In conclusion, the validity of the measurement model in this study was proven through the indicator

reliability, construct reliability, convergent validity, and discriminant validity. Next, we tested the structural model.

**Table 3.** Loadings and cross-loadings for the measurement model.

| Construct | Item | MPQ-TC | MPQ-EF | MPQ-EI | DL | SCT | SA | SMP |
|---|---|---|---|---|---|---|---|---|
| Techniques and Capabilities for Music Production (MPQ-TC) | **MPQ-TC2** | **0.380** | 0.466 | 0.423 | 0.343 | 0.144 | 0.413 | 0.121 |
| | **MPQ-TC3** | **0.279** | 0.565 | 0.607 | 0.292 | 0.227 | 0.381 | 0.161 |
| | **MPQ-TC4** | **0.508** | 0.599 | 0.660 | 0.394 | 0.397 | 0.327 | 0.047 |
| Emotions and Feelings for Music Production (MPQ-EF) | **MPQ-EF1** | 0.494 | **0.238** | 0.550 | 0.169 | 0.176 | 0.308 | 0.046 |
| | **MPQ-EF2** | 0.539 | **0.177** | 0.621 | 0.159 | 0.188 | 0.278 | −0.007 |
| | **MPQ-EF3** | 0.565 | **0.385** | 0.538 | 0.223 | 0.075 | 0.261 | 0.020 |
| | **MPQ-EF4** | 0.521 | **0.400** | 0.477 | 0.056 | 0.227 | 0.291 | −0.050 |
| External Interacting Factors Affecting Music Production (MPQ-EI) | **MPQ-EI1** | 0.579 | 0.545 | **0.537** | 0.377 | 0.274 | 0.299 | 0.045 |
| | **MPQ-EI2** | 0.491 | 0.542 | **0.382** | 0.180 | 0.154 | 0.216 | 0.035 |
| | **MPQ-EI4** | 0.479 | 0.400 | **0.349** | 0.502 | 0.224 | 0.304 | 0.114 |
| Deep Learning (DL) | **DL1** | 0.253 | 0.033 | 0.334 | **0.788** | 0.196 | 0.262 | 0.267 |
| | **DL2** | 0.251 | 0.085 | 0.288 | **0.805** | 0.149 | 0.278 | 0.179 |
| | **DL3** | 0.290 | 0.010 | 0.230 | **0.818** | 0.101 | 0.271 | 0.184 |
| | **DL5** | 0.365 | 0.216 | 0.461 | **0.813** | 0.382 | 0.267 | 0.079 |
| | **DL6** | 0.441 | 0.290 | 0.431 | **0.802** | 0.415 | 0.308 | 0.130 |
| Switching Costs of Technology Usage (SCT) | **SCT1** | 0.246 | 0.164 | 0.226 | 0.159 | **0.814** | 0.325 | 0.490 |
| | **SCT2** | 0.258 | 0.161 | 0.240 | 0.417 | **0.871** | 0.460 | 0.442 |
| | **SCT3** | 0.234 | 0.180 | 0.239 | 0.143 | **0.759** | 0.392 | 0.212 |
| | **SCT4** | 0.317 | 0.145 | 0.239 | 0.359 | **0.870** | 0.241 | 0.190 |
| Showing Appropriateness (SA) | **SA1** | 0.355 | 0.313 | 0.363 | 0.334 | 0.384 | **0.921** | 0.471 |
| | **SA2** | 0.438 | 0.318 | 0.287 | 0.310 | 0.402 | **0.934** | 0.525 |
| Satisfactions of Music Products (SMP) | **SMP1** | 0.062 | 0.099 | 0.065 | 0.043 | 0.219 | 0.370 | **0.791** |
| | **SMP2** | 0.172 | −0.022 | 0.106 | 0.270 | 0.385 | 0.554 | **0.869** |
| | **SMP3** | 0.024 | −0.068 | 0.008 | 0.145 | 0.378 | 0.393 | **0.860** |

Notes: The figures in bold represent the cross-loadings for the measurement model. MPQ-TC, MPQ-EF, and MPQ-EI constructs are formative indicators with loading weights above 0.2.

**Table 4.** Correlation matrix, composite reliability (CR), and square root of the average variances extracted (AVEs).

| Construct | CR | MPQ-TC | MPQ-EF | MPQ-EI | DL | SCT | SA | SMP |
|---|---|---|---|---|---|---|---|---|
| MPQ-TC | - | - | | | | | | |
| MPQ-EF | - | | - | | | | | |
| MPQ-EI | - | | | - | | | | |
| DL | **0.902** | | | | **0.648** | | | |
| SCT | **0.898** | | | | 0.331 | **0.688** | | |
| SA | **0.925** | | | | 0.347 | 0.424 | **0.860** | |
| SMP | **0.879** | | | | 0.201 | 0.399 | 0.538 | **0.707** |

Note: The bold figures represent the square roots of AVEs. (1) The first column features CR (composite reliability). (2) Diagonal elements are the square root of the average variance extracted (AVE). (3) Off-diagonal elements are correlations.

## 5.2. Structural Model

To evaluate the structural model, this study followed Hair's five-step approach [72]: (1) collinearity assessment, (2) structural model path coefficients, (3) coefficient of determination ($R^2$ value), (4) effect size $f^2$, and (5) predictive relevance $Q^2$ and blindfolding. This study's results suggest minimal collinearity among the constructs. The highest VIF (variance inflation factor) among the explanatory variables is 3.563, indicating a minimal correlation between predictors (i.e., independent variables) in the structural model. To empirically assess the hypotheses postulated in Section 3, this study examined the level of significance among the path coefficients (2) by means of a bootstrapping technique [67,69], with 5000 iterations of re-sampling and with each bootstrap sample constituted by the number of

observations (i.e., 105 cases). To achieve more conservative outcomes, the "no sign change" option was selected [72].

Figure 3 displays the estimated model (path coefficients, $R^2$ and $Q^2$), and Table 5 summarizes the results. In consideration of the $R^2$ values (3), all dependent variables present reasonable values. In addition, this study calculated the $f^2$ and $q^2$ effect sizes (4). Most of the values of the $f^2$ effect size are moderate and small, with the exception of "showing appropriateness (SA)" to the "satisfaction of music production (SMP)" (strong effects). Last, based on a blindfolding procedure, all $Q^2$ values are above zero, indicating a predictive model based on dependent variables.

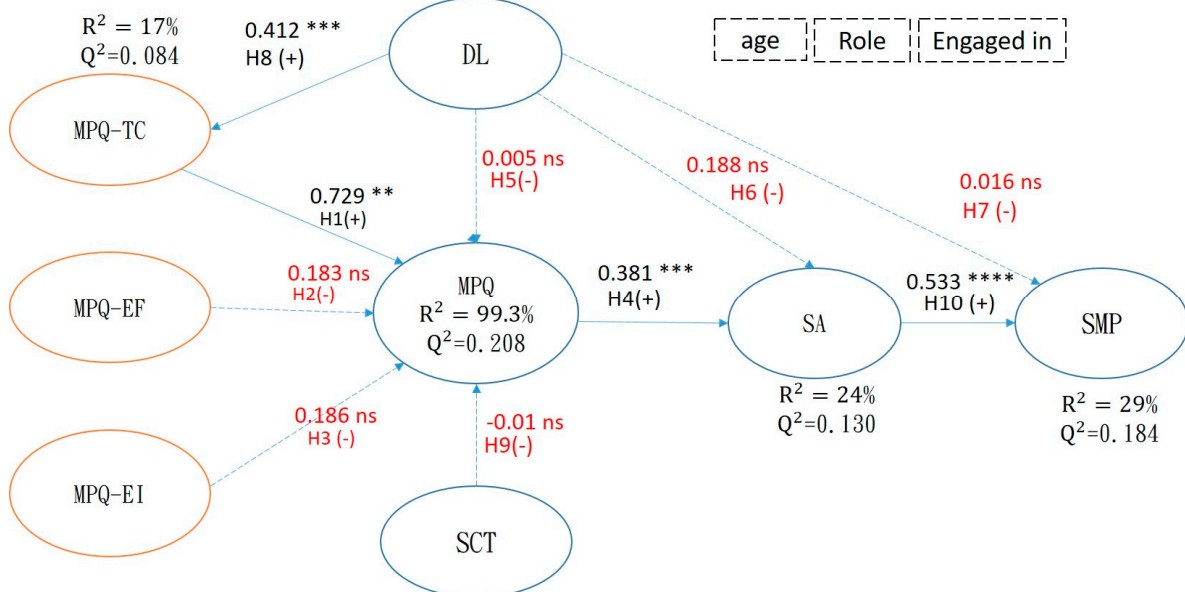

**Figure 3.** Estimated model. Note: This study followed Hair's five-step approach [67]; ns = non-significant. ** $|t| \geq 1.96$ at $p = 0.05$; *** $|t| \geq 2.57$ at $p = 0.01$ level; **** $|t| \geq 3.29$ at a $p = 0.001$ level.

**Table 5.** Significant testing results of the structural model path coefficients.

| Structural Path | Path Coefficient (*t*-Value) | Effect Size ($f^2$) | Effect Size ($q^2$) | 97.5% Confidence Interval | Conclusion |
|---|---|---|---|---|---|
| MPQ-TC → MPQ | 0.729 (2.446) ** | 31.242 | 0.118 | (−0.060; 1.091) | H1 supported |
| MPQ-EF → MPQ | 0.183 (0.6124) | 2.150 | 0.000 | (−0.339; 0.863) | H2 not supported |
| MPQ-EI → MPQ | 0.186(0.62998) | 1.981 | 0.000 | (−0.253; 0.909) | H3 not supported |
| MPQ → SA | 0.381 (3.195) *** | 0.157 | 0.314 | (0.174; 0.639) | H4 supported |
| SA → SMP | 0.533(5.438) **** | 0.351 | 0.189 | (0.351; 0.734) | H10 supported |
| MPQ → SCT | −0.01 (0.42) | 0.011 | 0.263 | (−0.042; 0.05)] | H9 not supported |
| MPQ → DL | 0.005 (0.2066) | 0.002 | 0.263 | (−0.044; 0.054) | H5 not supported |
| DL → SA | 0.188 (1.772) | 0.038 | 0.000 | (−0.030; 0.394) | H6 not supported |
| DL → SMP | 0.016 (0.1444) | 0.000 | 0.000 | (−0.177; 0.248) | H7 not supported |
| DL → MPQ-TC | 0.412 (3.2494) *** | 0.204 | 0.000 | (0.175; 0.662) | H8 supported |

Note: The values of $f^2$ and $q^2$ effects can be considered weak (0.02), moderate (0.15), and strong (0.35). Confidence level: ** $|t| \geq 1.96$ at a $p = 0.05$ level. *** $|t| \geq 2.57$ at a $p = 0.01$ level. **** $|t| \geq 3.29$ at a $p = 0.001$ level.

Figure 3 outlines the findings as follows. The conceptual model explains 99.3% of the variation in the "quality of music production" (MPQ). The "techniques and capabilities for music production" (MPQ-TC) ($\beta = 0.729$; $p < 0.05$) are statistically significant in explaining the "quality of music production" (MPQ). Thus, H1 is confirmed, whereas "emotions and feelings for music production" (MPQ-EF) (H2) and "external interacting factors affecting music production" (MPQ-EI) (H3) are not confirmed. The "quality of music production" (MPQ) ($\beta = 0.381$; $p < 0.01$) is statistically significant in explaining "showing appropriateness (SA)"; consequently, H4 is supported. The conceptual model explains 24% of

the variation in "showing appropriateness (SA)". "Showing appropriateness (SA)" ($\beta = 0.533$; $p < 0.001$) is statistically significant in explaining the "satisfaction of music products (SMP)"; consequently, H10 is supported.

The conceptual model explains 29% of the variation in the "satisfaction of music products (SMP)". "Deep learning (DL)" contributes significantly to the "techniques and capabilities for music production (MPQ-TC)" ($\beta = 0.412$; $p < 0.01$), which confirms H8. H5–7 are not supported due to statistical insignificance (DL → MPQ, DL → SA, DL → SMP). The conceptual model explains 17% of the variation in the "techniques and capabilities for music production (MPQ-TC)". The "switching costs of technology usage (SCT)" to "quality of music production (MPQ)" is not statistically significant. The conceptual model substantially explains the variation of all four dependent variables [69,73].

### 5.3. Testing Mediation Effects

Based on the guidelines from Hair [72], Preacher [74], and Nitzl [75], this study judged the significance of the mediating effects of "deep learning (DL)" and "switching costs of technology usage (SCT)", with the answers exhibited in Table 6.

The direct effect measures how the dependent variable changes when the independent variable changes, while the mediator variable remains unchanged. The indirect effect measures how the dependent variable changes when the independent variable remains unchanged, and the mediator variable changes. The total effects are equal to the sum of the direct and indirect effects. This study calculated variance accounted for (VAF) to determine how the indirect effects were related to the total effects [72]. The findings imply that "deep learning (DL)" and "switching costs of technology usage (SCT)" have partial mediating effects on the "quality of music production quality (MPQ)" and "showing appropriateness (SA)", thereby supporting H11a,b.

**Table 6.** Testing mediation by the bootstrapping approach.

| Effect of | Direct Effect (*t*-Value) | Indirect Effect (*t*-Value) | Total Effect | VAF (%) | Interpretation | Conclusion |
|---|---|---|---|---|---|---|
| MPQ → DL → SA | 0.096 (3.29) *** | 0.567 (6.484) **** | 0.663 (9.77) **** | 79.90% | Partial mediation | H11a supported |
| MPQ → SCT → SA | 0.126 (2.371) ** | 0.347 (2.54) ** | 0.473 (4.911) **** | 78.02% | Partial mediation | H11b supported |

Note: VAF = variance accounted for. The VAF >80% indicates full mediation; 20% ≤ VAF ≤ 80% shows partial mediation; VAF < 20% indicates no mediation. Ns = non-significant. Na = not applicable. ** |t| ≥ 1.96 at a $p = 0.05$ level. *** |t| ≥ 2.57 at a $p = 0.01$ level. **** |t| ≥ 3.29 at a $p = 0.001$ level.

## 6. Discussion

This study examined the essence of music to decide whether deep learning has a significant impact on the quality of music production. Prior deep learning projects produced satisfying artificial music that listeners could hardly distinguish from human creations. However, the statistical outcomes in this study do not answer whether deep learning can improve the quality of music production.

The results of this study indicate that the "techniques and capabilities for music production" have a significant impact on the "quality of music production", but the "emotions and feelings of music production staff" and the "external interacting factors of music production", do not have an association with the "quality of music production". This phenomenon is best explained by the fact that, with the adoption and application of technology, the barrier to music production is lower than before. Music production staff do not need to be limited to the rich emotions and unique personal traits that past scholars have emphasized as essential factors for musical creativity [65], as they only need to operate digital devices to generate satisfying music for the audience. Consequently, the managers in the industry require greater technical abilities than artistic sense and creativity for their music production staff. On the other hand, this phenomenon also reveals that the audience's demand for music is not determined at all by the emotions and feelings of the music production staff.

Music production usually has external interacting factors, such as professionals in different music fields and the different facilities involved. In live concerts, performers and environments might affect the music presentation [39]. Nevertheless, this study found that the "external interacting factors affecting music production" were not associated with the "quality of music production". This phenomenon is best explained by the fact that when listening to music in a live concert, the audience tends to determine the quality of music based on their overall feeling of the music and whether the music is in conformity with the situation and time of listening, not just on the external interacting factors affecting music production.

According to the findings of some studies, deep learning has diversified the music production process to achieve "showing appropriateness" of music and "customer satisfaction" [11,13]. However, this study found that "deep learning" did not affect the "quality of music production", "showing appropriateness", and the "satisfaction of the music product". Nevertheless, "deep learning" had a significant influence on the "techniques and capabilities for music production" and had a mediating effect on the "quality of the music production" and "showing appropriateness". This phenomenon is best explained by the fact that deep learning can improve the technical abilities of the music production staff. Due to the process of music production with a variety of professional staff and factors involved, music made by deep learning will not be able to show appropriateness to achieve customer satisfaction.

To change to new technologies, users will incur switching costs, such as time and money [63]. This study found that the "switching costs of technology usage" had no effect on the "quality of music production" but had a mediating effect on the "quality of the music production" and "showing appropriateness". This result interpreted the independency of "quality of the music production" on technology, even though technology assists in producing a variety of music. A good example to explain this occurrence is that classical music, without the adoption of technology in the past, is still enjoyed and appreciated by audiences at present because of its superior quality.

### 6.1. Limitations and Further Research

(1) The survey participants were chosen among staff in the music industry in Taiwan for this study. The sample does not represent the complete music industry in the world, as styles of music production and the acceptance of technology vary everywhere. The music production environment in Taiwan is very conservative, with ignorance of external stimuli. The adoption of new technology is also slow in Taiwan, as business managers always consider their rate of return on investment, as well as the shrinking market due to copyright piracy. In addition, the survey on customer satisfaction was limited to the front-end user behavior model, which did not cover all the economics of complicated buyer behavior. Future researchers can select a larger sample and apply more consumer behavior models to study the differences caused by different music market cultures and industrial relationships around the world.

(2) Fields of industry experience and education, such as popular or classical music, have influences on musical concepts and music production. The surveyed participants have various educational backgrounds, but the majority of them work in the field of popular music production. As people tend to think that the popular music industry relies heavily on digital technology for music production, future researchers can select a sample based on working and education backgrounds in all fields of music and study the implications of deep learning on the whole music industry, not just in the production area.

(3) Deep learning algorithms, theories, and concepts are currently in the early stages of development. There is not yet a complete application service or platform available for music production. In fact, very few music works are presently produced through deep learning. In addition, music production staff have a lower ability to understand and apply deep learning than information technology experts. As a result, the surveyed participants' cognitive gap on deep learning actually affected their responses to the questionnaire and the accuracy of this study. Future researchers



should return to the topic of deep learning when its development is more popular and mature in music production.

*6.2. Managerial Implications*

This study explored the impact of deep learning technology on music production industry. The managerial implications include:

(1) Position deep learning correctly in the industry: The invention of technology, because of market demand, can aid in the development and growth of the industry. To avoid the destruction caused by the improper usage of technology and waste of resources, business managers should deliberate on what, how, and when to utilize technology from the perspective of industrial improvement and social benefits. For example, P2P was meant for data sharing, but without proper rules, P2P caused copyright piracy issues and reduced the demand for quality music. This example suggests that technology should be suitably regulated to reduce any irreversible risks for the industry and the market. It remains too early to establish the correct direction for deep learning in music production. As this study uncovered a positive relationship between "deep learning" and the "techniques and ability for music production", the managerial indications of this work provide a milestone for business managers to set and achieve.

(2) Increase consumer awareness of the quality of music: Listeners are more likely to obtain low quality pirated music or products of kitsch culture through digital transmission, which negatively affects the perception of quality. As a result, music professionals do not demand high-quality music production but instead take advantage of digital technology to expedite the production process and save costs. Audio–visual products are commonly sacrificed by business budgets, which have precipitated a potential crisis in the industry. The managerial implication and driving force behind the development of music production in the future will be to build and increase consumer awareness of the quality of music.

(3) Enhance music production staff's acceptance of new technologies: Many music production staff are reluctant to adopt digitalization and new technologies due to the switching costs involved, such as money, time, attitudes, and learning curves. Deep learning is new but will definitely play a role in the future development of music production. As this study revealed, deep learning is significantly related to techniques and capabilities for music production. Thus, business managers should start to enhance their staff's acceptance of new technologies and strengthen their technical abilities to prevent outstanding music staff from being eliminated from the industry when the next big transaction arrives.

## 7. Conclusions

As technology can offer value to companies in several ways, many scholars highlighted the need to understand the path to competitive advantage and a model of sustainable development. The main outcome emerging from this paper has to do with understanding the value chain and sustainable development of the music production industry.

Grounded on past scholars' opinions on music production quality (MPQ) and artistic product satisfaction (SMP), this research fills the gaps in past research from the perspective of music as art through deep learning technology (DL) and technology conversion cost (SCT), focusing on the impact of deep learning technology on Taiwan's music production industry.

The results show that this model cannot explain all dependent variables (99.3% of the quality of music production, 24% of showing appropriateness, and 29% of satisfaction of music products). The major conclusions of this study are:

(1) Scholars' understanding of music production in the past and their current music cognition have gradually changed, partly because of the convenience of technology. Also, creative personnel have gradually reduced the cultivation of artistic literacy. On the other hand, due to changes in

market mechanisms and consumer perceptions, the overall atmosphere required by the audience is greater than the artwork itself.

(2) Deep learning has a significant impact on music production technology by improving the quality of music production. However, music is a unique expression of human creativity that can reflect life, thoughts, and emotions. Music is difficult to create through technologies such as computers, digitization, artificial intelligence, and deep learning.

(3) The results of this research remind technology developers that the development direction of deep learning technology is not to replace humans' unique artistic creativity, but it can lower the barriers for music producers to enter the production field (e.g., eMastered provides online mastering services through artificial intelligence [76]). Past musical works are of reference value. The application direction of deep learning should be focused on technical inheritance and improvement of music quality, so as to allow consumers to improve their music literacy as their main application direction.

(4) Although this research conclusion cannot cover all the situations in the various fields of music production in the world, we provide strategic warnings for the sustainable development of the music production industry. In addition to reviewing the education of music producers and listeners, this phenomenon also requires a more rigorous review of the use of science and technology in the music production industry and the direction of application development. Market mechanisms negatively determine the development of the industry or actively participate in it. The development of science and technology, as well as related policies, helps the industry play its maximum role, enabling science and technology to coexist with the industry from the best perspective and retaining humanity's most unique expressions of artistic creativity.

**Author Contributions:** S.-S.W. and H.-C.C. conceived and designed the research; H.-C.C. collected data and analyzed the data; S.-S.W. and H.-C.C. contributed to the progress of the research idea; H.-C.C. wrote the paper. All authors read and approved the final manuscript.

**Funding:** This research received no external funding.

**Conflicts of Interest:** The authors declare no conflicts of interest.

## Appendix A

**Table A1.** Survey Questionnaire.

| Construct | ID | Questions | References |
|---|---|---|---|
| Emotions and Feelings for Music Production (MPQ-EF) | MPQ-EF1 | Influenced by quality of experiencing life. | [2,77] |
| | MPQ-EF2 | Influenced by growing environments and situations. | [37,65] |
| | MPQ-EF3 | Influenced by personal traits. | [37,78–82] |
| | MPQ-EF4 | Influenced by the level of assimilation of emotions/feelings with external environments. | [2,65,80] |
| Techniques and Capabilities for Music Production (MPQ-TC) | MPQ-TC1 | Influenced by musical education background and knowledge of musical elements. | [2,36,80,83] |
| | MPQ-TC2 | Influenced by performance skill and the ability to adapt to changes. | [36,81,84] |
| | MPQ-TC3 | Influenced by the overall integration ability of the music production staff. | [2,36,79,80] |
| | MPQ-TC4 | Influenced by the technology used to assist in music production. | [81,85] |

**Table A1.** *Cont.*

| Construct | ID | Questions | References |
|---|---|---|---|
| External Interacting Factors Affecting Music Production (MPQ-EI) | MPQ-EI1 | Influenced by the external factors of musical creativity and performance. | [3,80] |
| | MPQ-EI2 | Influenced by interpersonal interactions with the outside world. | [84] |
| | MPQ-EI3 | Influenced by the interactions between the media and people through the use of technology. | [42] |
| | MPQ-EI4 | Influenced by the cooperation of music production staff in different professions. | [39,78] |
| Showing Appropriateness (SA) | SA1 | High-quality music can arouse a sense of "beauty", with internal feelings and emotions to satisfy spiritual needs. | [1,2,34,86] |
| | SA2 | Music has a direct effect on our souls through interactions with specific situations. | [20,33,42,87] |
| | SA3 | Appropriately presented music goes with the trend and one's lifestyle. | [3,88–91] |
| Satisfaction of Music Products (SMP) | SMP1 | High-quality music meets the expectations of return on investment. | [59,64] |
| | SMP2 | The quality of music determines the degree of satisfaction. | [57–59,92] |
| | SMP3 | High-quality music is pleasing and enjoyable. | [57,60–62] |
| Deep Learning in Music Characteristics (DL) | DL1 | Music is affected by tenuto and fermata. | [10–12,14,85] |
| | DL2 | Music is affected by piano and forte. | |
| | DL3 | Music is affected by chord. | |
| | DL4 | Music is affected by frequency. | |
| | DL5 | Music is affected by past performance methods. | |
| | DL6 | Music is affected by past tuning data for recording and mixing. | |
| | DL7 | Music is affected by cooperating video and peripheral interactive mechanisms. | |
| | DL8 | Music is affected by consumer preferences. | |
| Switching Costs of Technology Usage (SCT) | SCT1 | Switching to a new production technology can incur unpredictable economic losses. | [54–56] |
| | SCT2 | More time and effort is required when comparing the benefits that existing and new production technologies provide. | |
| | SCT3 | In order to use new technology effectively, I have to make more of an effort and take more time to acquire new skills and knowledge. | |
| | SCT4 | In order to change the method of production, I will go through a much more complex switching process. | |
| | SCT5 | The switching of new technology will waste previous equipment investments and cost more. | |
| | SCT6 | I will lose my partnership with other production staff by switching to new technology. | |
| | SCT7 | The use of new technology will negatively impact my original technical logic and previous product image. | |

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
