# Peer review of "Exploring the Role of Deep Learning Technology in the Sustainable Development of the Music Production Industry"

_sustainability, doi:10.3390/su12020625_

Round 1
Reviewer 1 Report
Dear Authors,
the paper is very interesting in both its premise and conclusions.
The methods employed for defining the research questions and for carrying out the statistical analysis are clearly described, although a brief explanation of what they are would be appreciated beside referncing the original work. (similarly tensorflow is mentioned as if everyone knows what it is).
I find that in some sections, in particular 1,2,6 and 7 which are more discursive, there is excessive redundancy in the language. Also, some words seems to be used inappropiately (e.g. "obsess" in line 402). This makes the paper hard to read, especially for the uninitiated.
I suggest you ask a native English speaker to proofread your paper if possible, so that those issues are quickly solved.
Reviewer 2 Report
The text describes field experiment on Deep Learning Artificial Intelligence influence on sustainability of music industry. Advantages of the article: 1) A research architecture for the problem was formulated. 2) Hypotheses about music industry was formulated. 3) Online survey was done in a pool of respondents involved in Taiwanese music industry. 4) Results was statistically investigated and conclusions has been drawn. Disadvantages of the article: 1) It is not clear if conclusions can pertain to other music industries/markets. Some of them can be pretty well separated i.g. as bolywood/holywood in film area. 2) The scientific soundness of the article can be improved by applying more literature references younger than 5 years.
